# Lower Limb Exoskeleton Sensors: State-of-the-Art

**DOI:** 10.3390/s22239091

**Published:** 2022-11-23

**Authors:** Slávka Neťuková, Martin Bejtic, Christiane Malá, Lucie Horáková, Patrik Kutílek, Jan Kauler, Radim Krupička

**Affiliations:** Faculty of Biomedical Engineering, Czech Technical University in Prague, 272 01 Kladno, Czech Republic

**Keywords:** exoskeletons, sensors, lower limbs, wearable robots, powered orthosis

## Abstract

Due to the ever-increasing proportion of older people in the total population and the growing awareness of the importance of protecting workers against physical overload during long-time hard work, the idea of supporting exoskeletons progressed from high-tech fiction to almost commercialized products within the last six decades. Sensors, as part of the perception layer, play a crucial role in enhancing the functionality of exoskeletons by providing as accurate real-time data as possible to generate reliable input data for the control layer. The result of the processed sensor data is the information about current limb position, movement intension, and needed support. With the help of this review article, we want to clarify which criteria for sensors used in exoskeletons are important and how standard sensor types, such as kinematic and kinetic sensors, are used in lower limb exoskeletons. We also want to outline the possibilities and limitations of special medical signal sensors detecting, e.g., brain or muscle signals to improve data perception at the human–machine interface. A topic-based literature and product research was done to gain the best possible overview of the newest developments, research results, and products in the field. The paper provides an extensive overview of sensor criteria that need to be considered for the use of sensors in exoskeletons, as well as a collection of sensors and their placement used in current exoskeleton products. Additionally, the article points out several types of sensors detecting physiological or environmental signals that might be beneficial for future exoskeleton developments.

## 1. Introduction

Wearable, powered exoskeletons can be classified into two main classes, medical and non-medical. The first class includes rehabilitative and assistive exoskeletons. The aim of these exoskeletons is to provide guided movement and facilitate labor-intensiveness by decreasing the load action on the wearer (rehabilitative) or physical support for daily living activities (assistive) [1]. The second class, non-medical exoskeletons are intended to be worn by healthy operators [2]. These exoskeletons are designed to provide an increased capability of carrying heavy loads with minimal effort, and to increase velocity, power, and endurance. Typical operators can be soldiers, disaster relief workers, fire fighters, or industry workers.

Exoskeleton applications are currently more concentrated in military and medical fields [3]. However, current achievements in exoskeleton technology are only the beginning, and a large-scale industry for practical application has not yet formed [4].

The exoskeleton control system generally includes three levels: the perception layer, the control layer, and the execution layer. The perception layer refers to the sensory system of the exoskeleton. The sensory system is responsible for obtaining data from the external environment (e.g., obstacles), the exoskeleton itself (e.g., exoskeleton state), the operator’s body (e.g., movement detection), and the operator–exoskeleton interaction. The control layer works with data obtained from the perception layer. Its task is to decide what kind of response should be used. Control mechanisms were reviewed by Yan et al. [5]. The function of the execution layer is to actuate the exoskeleton structures according to the results of the decision layer to make the exoskeleton carry out its action. Currently, electric, hydraulic, and pneumatic actuators are predominantly used in powered exoskeletons [6]. A new paradigm of “soft exoskeletons” has been increasingly adopted in the last decade. Soft exoskeletons (also called exosuits) attempt to replace big and rigid elements with soft, light, thin, and flexible ones [7], e.g., innovative textiles [8].

Current exoskeletons are required to be more and more advanced and accurate in shadowing the movement of the operators. This task not only requires the exoskeleton movement to be as close as possible to the operator, but it also involves early recognition of the operator’s intention. To meet these challenges, it is crucial to have an advanced sensory system.

The aim of the paper is to provide a connection between currently used sensors in powered lower-limb exoskeletons (hereon referred to as exoskeletons), an overview of other sensor possibilities for potential future use in exoskeletons, and the criteria which all sensors for exoskeletons should meet. The review can be used as a guideline for sensor type selection and help with decision making concerning new product developments.

The rest of this paper is organized as follows: related research is outlined in Section 1.1. A methodological review is described in Section 4. Sensor requirements are listed in Section 2. Sensing technologies for kinematics, kinetics, and physiological signals are detailed in Section 3. The application of sensors in lower-limb exoskeletons are detailed in Section 5. Section 6 concludes the paper and discusses the future direction of wearable sensing in lower limb exoskeletons.

### 1.1. Related Works

There is research centered on equipment utilized in biomechanics to study the structure and function of the human body that includes laboratory-based [9,10] sensors as well as wearable sensors [11,12]. Other publications reviewed wearable sensors in patient monitoring [13] and rehabilitation [14,15,16]. Ahmad et al. [17] pursued the application of an inertial measurement unit in several areas including medical rehabilitation.

Rukina et al. [18] solely focused on the usage of surface electromyography in the development and control of exoskeletons. They looked at the measurement, processing, and analysis of the muscle’s bioelectric activity with respect to exoskeleton control.

As far as lower limb exoskeletons are concerned, sensors were mentioned or listed in some publications [1,19,20,21,22,23]. However, none of this research was directly aimed at the employment of sensors in exoskeletons. These publications are rather general reviews on the topic of exoskeleton development.

Recently, Novak and Riener [24] provided an extensive overview of data fusion approaches that are used with a wearable robot or in similar conditions, e.g., a robot arm controlled by an EEG. Unlike their research, this article focuses on sensors rather than sensor fusion methods. Li et al. [25] reviewed control strategies for lower limb rehabilitation exoskeletons. Tucker et al. [26] summarized various types of sensors and focused on techniques for controlling portable active lower limb prosthetics and orthotic devices. Young and Ferris [27] reviewed common approaches in exoskeleton design, including actuators, energy supplies, control strategies, and materials. Some sensors were briefly mentioned in connection with movement perception and control methods [28,29] or assistive strategies [30]. Hussain et al. [31] examined various lower-limb robotic exoskeletons, concentrating on materials, manufacturing, and actuation.

Redkar et al. [32] focused on inertial measurement unit utilization in gait analysis and exoskeletons control strategies. Nevertheless, the relationship between these two topics is not strongly outlined in the article.

Unlike previous reviews, this article focuses on a wide range of sensors rather than specialized focus on purposefully selected sensors. Next, this review describes sensors broadly (focusing on sensors) rather than superficially (focusing on sensor fusion methods or control strategies).

## 2. Requirements for Sensor Characteristics

Batavia et al. [33] identified and prioritized a list of factors used by long-term users of assistive devices. This list was later refined by Lane et al. [34]. As sensors are the underlying technology to exoskeletons, sensors also need to meet these requirements. The eleven final criteria that were used to evaluate assistive devices along with a sensor analogy description are provided in Table 1. We excluded securability and portability criteria from the list. Further, the safety criterion has been justified in more detail. Securability should be guaranteed at the level of the entire exoskeleton, not just at the level of its individual components. Undoubtedly, portability is an essential and required feature of assistive and some empowering exoskeletons. The ability of the sensor to be moved or carried is an apparent feature as the sensors are worn by an operator and are used to capture his intent and movement. On the other hand, it is not expected that the sensor is independently able to relocate and operate in varied locations outside the exoskeleton. Sensors used in exoskeletons are currently input sensors that capture without interfering with the operator’s body. Although it is not expected that such sensors would harm the operator, it is necessary to keep in mind for future sensors and to assure the safety of the operator against harm.

Most of the requirements listed in Table 1 also apply to other exoskeleton classes (empowering and rehabilitation), see Table 2. Most of the requirements related to different exoskeleton classes are self-explanatory. However, it would be beneficial to comment on some of them. Acceptance is much more important for assistive exoskeletons when everyday personal use in public is anticipated than for empowering exoskeletons whose purpose greatly prevails over social adoption. The sensors cost (e.g., initial, repair) needs to be low to keep exoskeletons affordable for personal use, e.g., for assistive exoskeletons.

The next requirement for all standalone exoskeleton sensors is a low energy consumption. It is logical that exoskeleton energy efficiency needs to be improved to prolong the operation time [35].

## 3. Current Sensing Technologies

In this section, we provide information concerning sensors’ potential suitability or use in the design of lower-limb exoskeletons.

### 3.1. Kinematics Sensors

Among the most common kinematic wearable sensors used in exoskeletons are:Inertial measurement units (IMU), i.e., gyro-accelerometers [36],Electro-mechanical systems, i.e., electronic goniometers [37].

IMUs allows for the measurement of the rotational and translational variables of motion and for this reason, they have also been used in exoskeleton designs [38]. In addition to determining angles and segment positions, accelerometers are also used to identify gait phases based on instantaneous acceleration values. Information about the current gait phase is used in the control algorithms of the exoskeletons [35].

In the case of electro-mechanical systems, i.e., electronic goniometers, which were more recently developed, exoskeletons using electro-mechanical systems record kinematic parameters of motion on which the control of these robotic systems rely [39]. These sensors are the basis of measuring angular movements, especially for robot-aided assessment of lower extremity functions [40].

The limitations are as follows. Accelerometers occasionally need to be re-calibrated. Otherwise, there could be a signal offset due to changes in temperature, fluctuations in gain, or general mechanical wear [41]. Moreover, the sensor is susceptible to changes from precise attachment for reasons such as vibration (e.g., caused by movement of muscles during walking, impact on heel strike) which leads to a high frequency-based imprecision in the signal. The measured signal is influenced by gravity which requires additional compensation with signal processing. When an acceleration signal is integrated drift problems may occur [42]. Because the additional measurement of the earth’s magnetic field vector provides a second non-gravity affected reference, it may increase the measurement accuracy. Next, the assumption that the local coordinate axis of the sensors aligns with the joint coordinate axis can lead to errors over long periods [32].

### 3.2. Kinetics Sensors

A whole range of electronic sensors based on several distinctive methods have been designed. Among the most common and suitable for exoskeleton design are:Strain gauge sensors [43],Piezoelectric sensors [44].

These sensors can be used directly to measure force, or they can be modified and used to measure torque and/or pressure or tension. In the case of the lower-limb body movement measurements, force/torque sensors are placed on the shank or thigh, or directly on the joint (i.e., knee). To measure the latter, force and pressure sensors are used in the construction of medical aids [45]. Typically, they are elements that are directly incorporated into the mechanical design of the devices or systems.

Strain gauge sensors and piezoelectric sensors are a more expensive option and are usually used to measure forces transmitted by the exoskeleton segments. For this reason, several applications have already been used. Sensors are often used to determine the phases of the gait, walk ratio (step length/cadence), pose, etc. from the measured force or stress signal [46]. Foot-switches are utilized for gait event detection, for example the heel strike and heel off by switching to under the heel. In addition, sensors can determine the weight load which is then used as information for control algorithms to control exoskeleton actuators. Additionally, strain gauge and piezoelectric sensors have also been used to monitor the gait state and condition by measuring the ground reaction force (GRF) development and center of pressure development in stance phases [47]. The strain gauge and piezoelectric sensors have been also used to measure the pressure distribution on an exoskeleton [48].

The disadvantage of force-based sensors can be that placement for operators with abnormal gait is difficult [49]. For this reason, sensors are often installed directly into the construction of the exoskeletons outside the body segment. However, the accuracy of identifying gait events can be reduced [50].

### 3.3. Muscle Activity Sensors

Measuring muscle activity has several uses in determining the physical and mental state of the subject being measured. Since the electrical activity of muscles precedes movement, sensing muscle activity allows for estimating the operator’s movement prior to the movement occurring [51].

For the application of sensors used in exoskeleton design, the most commonly used are surface electromyography (EMG) sensors. EMG sensors are typically used to study the state and condition of specific muscles [52] and use the measured signal in actuator control algorithms [53,54]. In case the exoskeleton is used for robotic rehabilitation, the EMG data are measured and analyzed to describe the neuromuscular activity and interaction between the exoskeleton and the user [55]. Applying mathematical models, it is possible to estimate appropriate forces and movements of the lower limbs [53] and consequently, the exoskeleton can apply assistive movements.

Another type of sensor used in practice is the muscle pressure sensor [56]. The muscle pressure sensor has been tested on an actuated knee orthosis with artificial muscles. The user’s knee extension intention was estimated by the muscle stiffness, according to which the actuators were controlled [11,56].

Mechanomyography (MMG) is a measurement technique used to record muscle activity based on vibrations arising as an effect of the muscle fibers’ mechanical contractions. Various types of transducers, such as accelerometers, microphones, or laser distance sensors, can be used to convert mechanical vibrations to electrical signals [57,58]. It has been shown that an MMG measured by microphone sensors on different days is reliable and relates to changes in forces [59].

The disadvantage of EMG sensors is that the acquired signal is highly influenced by the location of the electrodes, skin tissue impedance (e.g., caused by sweat), and muscle size. As the signals pass through numerous tissue layers before they reach the skin surface, the signals are prone to cross-talk, interference, and noise [60] Further, the electrodes do not isolate and register the activity of a single muscle (e.g., controlling the particular movement), but all those that are located close to the surface covered by the electrode.

Various factors affect the nature of the signals recorded and therefore need to be considered when using MMG sensors. These include contact pressure, temperature, and positioning of the sensor [61,62,63,64,65]. These considerations are crucial and might need to be resolved prior to utilization in the exoskeleton.

There is another class of muscle activity sensors that focuses on changes to the mechanical properties of muscles, including muscle elastography sensors [66], muscle stretch sensors [67], muscle pressure sensors [68], and resonance muscle stiffness sensors. Muscle elastography sensors are based on ultrasound technique with the aim to non-invasively assess localized muscle stiffness. The muscle stretch sensors [67] placed on the measured muscle are usually made of conductive plastic material which changes its resistance depending on the amount it is stretched. The muscle pressure sensors [68] based on the piezo-resistance principle are used to measure the pressure from the muscle to which they are attached. The sensors allow for measuring the force exerted by the muscle or to sense [69] the deformation of the muscle [70]. In addition, muscle stiffness sensors based on the measurement of resonance signal changes have been tested [71]. The disadvantage of these sensors is that the mechanical sensing of muscle contractions is slower than the EMG signal measurement.

When myographic signals are used in the exoskeletons field, one of the essential areas of optimization is the number and placement of the sensors used [72]. To enforce endeavors to improve sensing quality, multimodal approaches as well as new sensor configurations are looked for. One possible approach is combining sensors. An example of coupling a microphone and an accelerometer [73] or EMG and MMG [74,75,76,77] has been shown. Zhang et al. [77] showed that an introduction of MMG signals might significantly improve the performance of an EMG-pattern recognition-based prosthetic control.

### 3.4. Brain Activity Sensors

Brain activity measurements can be divided into invasive and noninvasive according to the acquisition technique. For robotic systems, researchers prefer noninvasive measurement by using an electroencephalogram (EEG) signal. EEG signals are electric brain signals obtained by placing electrodes on the scalp to measure the summation of neuron potentials. Usually, the international 10–20 system is followed.

An EEG is the most common tool in brain–computer interfaces, although more recently, functional magnetic resonance imaging (fMRI) [78], functional near infrared spectroscopy (fNIRS) [79], magnetoencephalography (MEG) [80] and functional transcranial Doppler ultrasonography [81,82,83] have been considered.

An EEG provides relatively poor spatial resolution, but fine temporal resolution [84,85,86,87]. Since an EEG captures only the electrical field and since the brain is a high energy-demanding organ and neuronal activation correlates with increases in cerebral blood flow and volume, accessory analysis of blood-oxygen level-dependent (BOLD) activity may improve brain activity assessment performance. BOLD activity is typically captured with an fMRI [88]. Alternatively, the BOLD can be acquired safely and non-invasively by fNIRS [79]. A recent study indicated that fNIRS is unable to adequately offer acceptable performances on its own [89]. However, detecting and quantifying brain activity signals to discern the user’s movement intention can be boosted by the integration of fNIRS with an EEG [89].

MEG has the advantage of recording brain activity across the whole scalp while maintaining much higher spatial and temporal resolution. Compared to an EEG, MEG allows for detecting higher frequencies such as magnetic fields which are less attenuated by the head bone and tissue as compared to electric fields [90]. A MEG recording constitutes another potential approach when combined with an EEG, as it adds complementary information to the EEG signals [91].

EEG-based brain–computer interfaces have been discussed in excellent reviews published by Rashid et al. [92] and Saha et al. [93]

The disadvantage of an EEG is that it needs gel or saline liquid to reduce the impedance of skin–electrode contact [94]. The problem with the liquid medium is that it dries with time. However, currently there are some dry electrodes that have been developed which might solve this problem [95,96].

Although BOLD acquisition by an fMRI can be beneficial in brain activity assessment, the size and cost of the device makes it unsuitable for application in the exoskeleton field.

Though not portable, MEG-based brain activity monitors are relevant for use as rehabilitation exoskeletons rather than empowering or assistive.

### 3.5. Other Physiological Signal Sensors

Beside sensing kinetics, kinematics, and muscle and brain activity, other physiological signals can be measured. There are several sensors within a range of physiological indicators, including heart rate sensors, respiratory rate sensors, blood pressure sensors, pulse oximetry sensors, acoustic sensors, temperature sensors, galvanic skin response sensors, and perspiration sensors. The list and properties of these suitable wearable sensors are described in detail in [97].

For lower-limb exoskeletons in practice, physiological signal sensors are not usually used when compared to the above-mentioned sensors. Of course, a number of studies have been carried out to evaluate the physiological data measured during the use of exoskeletons; however, physiological data sensors were not used in any of the cases as a component of the lower limb exoskeleton [98]. Only preliminary designs of lower limb exoskeletons contained the most basic sensors implemented into exoskeleton construction, such as temperature and perspiration (i.e., humidity) sensors, were introduced [99,100].

## 4. Methodology

The aim of this article was not to conduct a systematic review of the literature with precise search criteria, but instead to propose a thematic review based on the most recent articles and those already known by the authors.

Two databases—IEEE Xplore (Institute of Electrical and Electronics Engineers and Institution of Engineering and Technology) and ScienceDirect (Elsevier)—were used for literature search. A combination of keywords, such as exoskeleton, sensors, wearable, and kin—were used as search terms. Publications from 2005–2022 were preferred; however, this range was extended in some cases.

In order to find resources which provided more explicit information on lower-limb exoskeleton sensors, we further traversed surveys focusing on lower-limbs exoskeletons [1,5,39,101,102]. However, none of the articles were focused on lower-limb exoskeleton sensors, or the current state of their use and perspectives.

## 5. Results

Angle sensors are usually mounted at the hip, knee, and ankle to measure the joint’s movement angle. Angular acceleration sensors are usually installed on the thigh and shank to obtain its angle, angular velocity, and acceleration. Force and pressure insole sensors are usually used to detect gait events and phases of the gait cycle [103,104,105,106]. Beside insole sensors, PERCRO BE [105] uses sensors placed on the trunk which are employed for detecting its orientation with respect to gravity. Strain gauges placed on the structure of the exoskeleton shank are used to measure the bending moment of the shank as well as the vertical force in the exoskeleton leg [107,108]. A different approach in signal measurement can be seen in the Berkeley lower extremity exoskeleton (BLEEX) that does not need any direct measurements from the operator or the operator-exoskeleton interface [109]. Rather, it uses sensors built-in into the exoskeleton structures [110].

The Naval Aeronautical Engineering Institute Exoskeleton Suit (NAEIES) [111] does not use any sensors placed in the area of the lower extremities (neither on the lower limbs nor on the exoskeleton). The movement of the lower limb exoskeleton is based on the idea that the forearm has a similar motion trajectory with the knee joint during gait. The forearm motion is measured by a potentiometer and afterwards used as the control signal for the knee joint.

The Exoskeleton HAL-5 Type-B employs an EMG sensor on the thigh together with an angle sensor at the joints and pressure sensors under the feet. Only two empowering exoskeletons rely solely on sensing of muscle activity [112,113].

For a summary and comparison of sensors’ placement, as described above, see Table 3.

## 6. Discussion

The question addressed in this review was: what current sensors are employed in lower-limb exoskeletons? Based on the analysis of the current state, sensors can be grouped into the following three basic groups, which are discussed separately:Movement activity sensors and sensors measuring the state of the musculoskeletal system,Sensors measuring physiological and other biomedical data,Sensors for measuring the physical characteristics of the state of the exoskeleton and the environment.

### 6.1. Movement Activity Sensors and Sensors Measuring the State of the Musculoskeletal System

Various sensors and sensor combinations are used in exoskeletons to provide signals that allow human movement recognition. Force sensors under the feet tend to be the most used kinetic sensors. Force sensors are often used in combination with angle sensors which are the most used kinematics sensors. Overall, kinematics and kinetics sensors are more commonly used than biological signal sensors.

The ability to recognize human movement accurately in real-time is crucial for exoskeleton performance. Movement modeling and control algorithms are related to the proper design and function of the exoskeleton. Employed sensors are essential as these tasks rely on input signals (e.g., accuracy, synchronization). Thus, the consideration of some sensor features should be mentioned. Especially the number of sensors, data redundancy and complementarity, and sensor placement may be emphasized.

Although sensors attached to the foot provides satisfactory results in gait phase recognition for normal walking [132], sensor information from other parts of the body is necessary for identifying more challenging movements. The use of more sensors can achieve improved accuracy in movement recognition. On the other hand, a greater number of sensors imply a greater number of acquired signals, which may lead to increased complexity of movement information processing, computational demands, and thus time delay signal processing (and movement identification). As wearable exoskeletons are battery-powered devices, the number of sensors coheres strongly with energy efficiency. With respect to the exoskeleton’s purpose, the number of sensors should be considered to attain a balance of accuracy and complexity.

In some cases, a variability of interest can be obtained by different methods. For example, the joint angle can be measured by resistive potentiometers, optical encoders, etc. In such cases, comparison and assessment of various methods should be performed considering daily usage and the operability of the sensor. Rueterbories et al. [132] pointed out that sensor positioning in gait phase detection seems less critical than placing the sensor on nearly any combination of foot, shank, thigh, and trunk of one or both legs being possible with appropriate signal processing. Moreover, the utilization of multiple kinematic sensors allows one to estimate the kinematics of movement [43,133,134,135]. However, sufficient reliability cannot be obtained by solely relying on accelerometers [136]. To date, a consensus has not been reached on which are the best suited sensors for exoskeleton control. Thus, additional effort is needed to determine the optimal combination and placement of sensor for gait and other movement tasks. Furthermore, sensor compatibility relating to which movement identification task is performed should be analyzed.

Although accelerometers are the most common wearable sensors that are used in gait analysis [132,137], their usage in exoskeletons is negligible. This might be caused by the impossibility of determining the direction of movement from acceleration. Thus, analysis and interpretation of gait data are perhaps easiest when the walking pattern is cyclical, in a straight line, and the walking speed is steady state [138]. These conditions cannot be expected in assistive and empowering exoskeletons.

Inspired from other fields of biomedical engineering where wearable sensors are utilized, e.g., prosthesis control, gait analysis, there are a number of sensors that could be promising in lower limb exoskeletons but are not employed yet. The ultrasonic imaging of dynamic muscle activity, which is known as sonomyography (SMG), could be investigated. In comparison with an EMG, a SMG is able to acquire signals from the muscles at different depths and avoid cross-talk from adjacent muscles. A SMG is less influenced by disturbance from fat or surface skin impedance, as shown in our previous studies [139,140]. Moreover, as a SMG could provide a safe and non-invasive approach to track superficial muscle activity [141], it seems to be a suitable method for sensing operator movement for exoskeleton control. A SMG has already been proposed for controlling upper-limb-powered prostheses [142]. Relating to gait analysis, there is insufficient research focused on SMG in lower-limb muscles [141,143].

An exoskeleton should identify and adequately respond to operator fatigue via a level of provided assistance. Although research in the field of muscle fatigue detection shows promising results, there is very little research carried out on the prediction and detection of fatigue while wearing a stand-alone exoskeleton [144]. Various sensors that are already used in exoskeletons provide concerned signals, e.g., joint angles [145,146], EMG [147,148]. Besides MMG [149] and SMG sensors, [146], accelerometers [147] or Near Infrared Spectroscopy (NIRS) sensors [148] can be exerted to acquire source signals for fatigue assessment.

### 6.2. Sensors Measuring Physiological and Other Biomedical Data

Physiological state sensors are not used in exoskeletons of the lower extremities, but they are sometimes used as a complementary means of monitoring the subject’s state of health. Heart rate sensors and pulse oximetry sensors are used to monitor the patient’s state while using the exoskeleton to improve functional gain and fitness [100]. Pulse oximetry sensors, respiratory rate sensors, and heart rate sensors were used to evaluate the effects of robotic knee exoskeleton on human energy expenditure [150]. Blood pressure, pulse, and electrocardiography sensors were used to evaluate the safety and tolerance of use of the ReWalk™ exoskeleton ambulation system.

Electroneurography (ENG) has shown promising results in upper-limb prostheses [151,152]. In the case of ENG, wearers can benefit from the possibility of conveying sensation back to the wearer.

Beside rehabilitation [153], EEG has found an application in wheelchair [154] and hand orthosis [155] control. It suggests the potential of being used in empowering and assistive exoskeletons as well.

### 6.3. Sensors for Measuring the Physical Characteristics of the State of the Exoskeleton and the Environment

The exoskeletons do not seem to be fully mature to be adopted for strenuous and non-programmed tasks. To deal with this shortcoming, different aspects need to be analyzed and solutions developed. For example, knowledge regarding the environment seems to be valuable for exoskeleton control. Environmental elements impede an operator’s movement, e.g., obstacle crossing or circumventing, and force the operator to unexpectedly change movement direction or to perform a compensatory movement to negotiate. Moreover, the environment has great influence on the stability, balance, and energy consumption of the operator [156]. It seems to be worth discerning environment elements by an exoskeleton [27]. Gyroscopes together with infrared sensors were used to estimate the slope in a powered prosthesis [157]. Sonar sensors and digital cameras were applied in wheelchairs to detect obstacles [158]. In line with obstacles scanners, exoskeletons could be equipped with a unit to provide a haptic signal to inform operators about obstacles and congestion.

There is a rapidly growing and advancing field of textile sensors. Fabrics which are equipped with sensing features are called smart fabric sensors. This class of sensors can be sensitive to multiple physical and chemical stimuli, including temperature, pressure, force, and electrical current, among others [159,160]. Utilization of such technologies in an exoskeleton’s perception layer can improve learnability, comfort/acceptance, and operability of exoskeletons sensors and consequently exoskeletons too.

## 7. Conclusions

Sensors are one of the most important parts in exoskeletons, as the data collected by them decide the start, the intensity, and the end of the support given by mechanical parts, and with this decide the quality of the whole system. There is a wide range of technical and not-technical criteria which sensors need to fulfill, to make the product not only technically secured but also acceptable and useful for a possible customer or patient. Next to general criteria for sensors such as reliability, safety, and durability, more special criteria for the use by end customers or patients in everyday life are given. The most important, mentioned here, are maintenance and repair requirements, the effectiveness of capturing sensed values without disturbing operators’ movement, and comfort of wearing, as the user will probably wear the exoskeleton a few hours a day.

The most widely used types of sensors in currently available exoskeletons are kinetic and kinematic sensors placed under the foot and in the joint regions (ankle, knee, hip). They are able to reliably detect gait phase, start and end of the movement, and direction of the movement. The use of biosignal sensors, such as EEG, fNIRS, EMG, MMG, EMG, and similar, in data collection for exoskeletons can be beneficial from the side of data collection, as it gives a deeper insight to movement intensions. On the other hand, these sensors currently have the limitation that they need a direct electrode–body interface, which does not meet the above-mentioned criteria of low maintenance and wear comfort. Further biomedical research is necessary to develop measurement methods for the detection of these signals without direct body contact electrodes. Then, their usage in exoskeletons will be possible and will improve the performance of them. Sensors detecting environmental data such as temperature, chemicals, force, etc. are technically ready to use and with the development of integrating such sensors in fabrics, they might be a further option for data collection in exoskeletons. Even if they might not be necessary for basic functions of the supporting system, they will be able to provide additional information to provide a more complex dataset about inner and outer conditions for exoskeleton tasks.

Sensors for exoskeleton developments should be chosen carefully, checking the criteria necessary for the intended usage. Developers can choose from a wide range of already existing sensor types. With ongoing research in the field of medical technology, new sensors could emerge soon, extending the range of detectable signals and improving the performance of the exoskeleton.

## Figures and Tables

**Table 1 sensors-22-09091-t001:** Assistive devices requirements and how they relate to exoskeletons sensors.

Criterion	Definition [34]	Exoskeletons Sensor Requirements
Effectiveness	How much the device improves one’s living situation, enhances functional capability and independence.	Sensed values need to be captured in such a way (e.g., kind of signal, frequency) that the exoskeleton can maintain the operator’s quality of movement.
Affordability	The extent to which a person can purchase, maintain, and repair a device without financial hardship.	Sensors on the exoskeleton should not be too financially demanding to remain financially viable.
Reliability	The degree to which a device is dependable, consistent, and predictable in its performance and level of accuracy under reasonable use.	The sensors should output consistent and predictable values despite environmental conditions such as water droplets, moisture, and sweat.
Portability	The influence of the device’s size and weight on the user’s ability to move, carry, relocate, and operate it in varied locations.	The sensors should be portable, small, and energy-saving.
Durability	The extent to which a device delivers continued operation for an extended period of time.	The expected useful lifetime of the sensor (despite environmental impurities, e.g., dust).
Securability	How well a consumer believes a device affords physical control and can be secured from theft or vandalism.	The sensors should be suitably attached to the structure of the exoskeleton.
Safety	The physical security a device affords the user, and how well it protects the user, care provider, or family member from potential harm, bodily injury, or infection.	The protection of the operator from potential harm, injury, infection, etc.
Learnability	The perspective of the device’s ease of assembly, initial learning requirements, and time and effort to master use.	Initial learning difficulty to use (e.g., attach, detach) the sensors.
Comfort/Acceptance	The extent to which a user feels physically comfortable with the device and does not experience pain or discomfort with use; how aesthetically appealing the user finds the device and the user’s psychological comfort when using it in private or public.	The fit, appearance does not cause the user to feel stigmatized while using the device.
Maintenance/Reparability	The degree to which the device is easy to maintain and repair (either by the consumer, a local repair shop, or a supplier).	The same as definition.
Operability	The extent to which the device is easy to use, adaptable and flexible, and affords easy access to controls and displays.	The perspective of the sensor being easily fastened in the intended position (with respect to the operator’s body) including a certain tolerance to the exact position.

**Table 2 sensors-22-09091-t002:** Applicability of requirements to exoskeleton classes.

Criterion	Required in Assistive Exoskeletons	Required in Empowering Exoskeletons
Effectiveness	Yes	Yes
Affordability	Yes	No
Reliability	Yes	Yes
Durability	Yes	Yes
Learnability	Yes	Yes
Comfort/Acceptance	Yes	No
Maintenance/Reparability	Yes	Yes
Operability	Yes	Yes

**Table 3 sensors-22-09091-t003:** Placement of sensors employed in lower limbs exoskeletons. A—acceleration sensor, Po—potentiometer, G—gyroscope, E—encoders F—force sensor, S—strain gauge, FS—foot switch, P—pressure sensor.

	Kinematics Sensors				Kinetics				Muscles Activity
Exoskeleton	Hip	Knee	Ankle	Other	Thigh	Shank	Foot	Other	
BLEEX [103,109,110,114]	-	-	-	A and E built-in in exoskeleton structures	-	-	FS	-	-
MIT exoskeleton [107,108]	Po	Po	-	-	F	S	-	-	-
Agri-Robot [115]	Po	Po	Po	Po on shoulder and elbow; G (placement is not published)	-	-	-	F (placement is not published)	-
PERCRO BE [105]	-	-	-	A on the trunk	-	-	F	F on the trunk, hands	-
NAEIES [111]	-	-	-	Po on forearm	-	-	-	-	-
HEXAR- CR50 [106]	-	-	-	-	F	-	F	F in the waist harness	-
Nursing exoskeleton [112,116]	-	-	-	-	-	-	-	-	MSS above the knees; EMG sensor on the upper arms and back above the hip
Hanyang University exoskeleton [113]	-	-	-	-	-	-	-	-	MSS on the thigh above the knees, MSS on the calf below the knee
WPAL (walking power assist leg) [117]	E	E	-	-	F	F	F	-	-
IHMC mobility assist exoskeleton [104,118] and Mina [119]	-	-	-	position sensors on the actuators	-	-	P	F on the actuators	-
ReWalk [120]	-	-	-	tilt sensor on the torso	-	-	-	-	-
Rex	-	-	-	-	-	-	-	-	-
ELegs	-	-	-	-	-	-	-	-	-
HAL-5 Type-B		Po	Po	-	-	-	P	-	EMG sensors on thigh
HAL-5 Type-C		Po	Po	-	-	-	P	-	-
AUSTIN [121]	E	-	-	-	-	-	-	-	-
MindWalker [122]	E	E	E	-	-	-	-	-	-
By G. Belforte [123]	Po	Po	-	-	-	-	-	-	-
Indego/Wanderbilt [124]	E	E	-	-	-	-	-	-	-
BioMot [125]	E	E	E	-	-	-	-	-	-
C Brace [126]	-	angle sensor, velocity sensor	-	-	-	-	-	ankle moment sensor	-
Exo-Lite [127]	-	-	-	angle sensor	-	-	P	-	-
H-MEX	-	-	-	-	-	-	F	-	-
Hank [128]	-	-	-	-	F	F	F	-	-
Keeogo [129]	-	-	-	knee, hip, thigh, no definition	-	-	-	-	-
KIT-EXO-1 [127]	-	-	-	-	F	-	F	-	-
INDEGO [130]	angle sensor	angle sensor	-	A on the thigh, tilt sensor on the trunk	-	-	-	-	-
ReStore [131]	-	-	-	inertial sensor on calf	-	F	-	-	-
ExoRoboWalker	Po	Po	Po	-	-	-	P	-	-

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
