# Peer review of "Lower Limb Exoskeleton Sensors: State-of-the-Art"

_sensors, 2022, doi:10.3390/s22239091_

Round 1

Reviewer 1 Report

This manuscript describes Lower Limb Exoskeleton Sensors: State-of-the-Art. Though some findings are interesting, there are several limitations/flaws which should be solved before it is considered for publication. Please see the list of my comments below.

Major:

1.     The abstract should focus on the findings of the review rather than on the background description.

2.     In the methodology part, I did not see the query string and exclusion criteria and therefore could not assess the validity of this review. I think this manuscript would be appropriate for PRISMA formatting.

3.     In Section 3, the reason why you chose these requirements for sensor characteristics seems not to be described using strong evidence.

4.     Please check that the title of Table 2, “Most of the requirements related to different exoskeleton classes are self-explanatory” is correct, as it seems somewhat inappropriate here.

Minor:

I could see considerable number of language issues and formatting errors (including reference). Please correct them. Indication of those are not a task of the reviewer. Please consider editing the manuscript before resubmission. For example:

1.     In line 7 and 8, “The number of elderly populations is rapidly increasing. Majority of elderly people face difficulties while walking.” 

2.     In line 74 and 75, “this article focuses rather on sensors rather than sensor fusion methods.” 

3.     In line 86 and 87, “Finally, briefly mention the main aim of the work and highlight the principal conclusions.” 

Author Response

Reply: Dear Reviewer #1. Thank you for all your comments on our manuscript. We appreciate your comments and revised the manuscript accordingly. Please note we also accommodate comments from Reviewer #2. Below you will find a detailed point-by-point reply to your comments, indicating the changes made to the manuscript.

This manuscript describes Lower Limb Exoskeleton Sensors: State-of-the-Art. Though some findings are interesting, there are several limitations/flaws which should be solved before it is considered for publication. Please see the list of my comments below.

Major:

  1.     The abstract should focus on the findings of the review rather than on the background description.

Reply: We updated the abstract according to this comment: we summarized the most utilized sensors and their location. 

  1.     In the methodology part, I did not see the query string and exclusion criteria and therefore could not assess the validity of this review. I think this manuscript would be appropriate for PRISMA formatting.

Reply:  In the Methodology section it is stated that a combination of keywords, such as exoskeleton, sensors, wearable, kinematics, kinetics, electromyography, surface electromyography, and muscle stiffness were used as search terms. Neither we include query string nor employ PRISMA structure and formatting style as usual in systematic review. Rather, we follow practices that are followed in other state of the art reviews published in prestigious journals, e.g. [R1][R2].

[R1] Young AJ, Ferris DP. State of the Art and Future Directions for Lower Limb Robotic Exoskeletons. IEEE Trans Neural Syst Rehabil Eng. 2017 Feb;25(2):171-182. doi: 10.1109/TNSRE.2016.2521160

[R2] M. Xiloyannis et al., Soft Robotic Suits: State of the Art, Core Technologies, and Open Challenges, IEEE Transactions on Robotics, vol. 38, no. 3, pp. 1343-1362, June 2022, doi: 10.1109/TRO.2021.3084466.

  1.     In Section 3, the reason why you chose these requirements for sensor characteristics seems not to be described using strong evidence.

Reply:  The selection of requirements is based on highly cited articles, namely [33] and [34]. In addition to our manuscript, these articles are adopted in other works in the field, e.g. [R1], [R2]. It would be very kind of you to reassess your opinion.

[33] Batavia, A.I.; Hammer, G.S. Toward the development of consumer-based criteria for the evaluation of assistive devices. J. Rehabil. Res. Dev. 1990, 27, 425–436, doi:10.1682/jrrd.1990.10.0425.

[34] Scherer, M.J.; Lane, J.P. Assessing consumer profiles of 'ideal' assistive technologies in ten categories: an integration of quan-titative and qualitative methods. Disabil. Rehabil. 1997, 19, 528–535, doi:10.3109/09638289709166046.

[R1] Gonzalez, A., Garcia, L., Kilby, J. et al. Robotic devices for paediatric rehabilitation: a review of design features. BioMed Eng OnLine 20, 89 (2021). https://doi.org/10.1186/s12938-021-00920-5

[R2] Veale AJ, Xie SQ. Towards compliant and wearable robotic orthoses: A review of current and emerging actuator technologies. Med Eng Phys. 2016 Apr;38(4):317-25. doi: 10.1016/j.medengphy.2016.01.010

  1.     Please check that the title of Table 2, “Most of the requirements related to different exoskeleton classes are self-explanatory” is correct, as it seems somewhat inappropriate here.

Reply: Thank you for this observation. We replaced the title of Table 2 with the appropriate one.

Minor:

I could see considerable number of language issues and formatting errors (including reference). Please correct them. Indication of those are not a task of the reviewer. Please consider editing the manuscript before resubmission. For example:

  1.     In line 7 and 8, “The number of elderly populations is rapidly increasing. Majority of elderly people face difficulties while walking.” 

Reply: The manuscript was re-edited by the native speaker to remove language issues.

  1.     In line 74 and 75, “this article focuses rather on sensors rather than sensor fusion methods.” 

Reply:  The manuscript was re-edited by the native speaker to avoid such issues.

  1.     In line 86 and 87, “Finally, briefly mention the main aim of the work and highlight the principal conclusions.” 

Reply:  We definitely removed this part. It was a leftover document template. We are sorry for this mistake.

Reviewer 2 Report

Dear authors

It is a complete work, which presents valuable results for the scientific community. It has satisfactory results that meet the objectives of the study.  

Here are some observations that could improve the content of the study. So that various technological areas understand the importance of your research.

Line 21 to 24 - You mention some medical uses, such as assistance and rehabilitation. Here you could put some references. 

Line 27 to 28 - You mention some non-medical uses, like in soldiers, firefighters, etc. Here you could put some references.

In the introduction you could mention the soft exoskeletons, and their advantages to be used in the support of the lower joints. You can be guided by the following article to further enrich your article:

"Pérez Vidal, A. F., Rumbo Morales, J. Y., Ortiz Torres, G., Sorcia Vázquez, F. D. J., Cruz Rojas, A., Brizuela Mendoza, J. A., & Rodríguez Cerda, J. C. (2021, July). Soft exoskeletons: development, requirements, and challenges of the last decade. In Actuators (Vol. 10, No. 7, p. 166). MDPI".

Table 1. Each paragraph must begin with a capital letter. You should also create a space between rows to differentiate them, because they get confused.

Line 141 to 142 - You must start with a capital letter.

Line 153 and 187. It's confusing how you put this "Limitations" subtitle. Better put it as part of the paragraph. For example, "Limitations are..."

. Table 2 should be sent to an annex.

The article can be accepted after having made the pertinent observations.

Author Response

Reply: Dear Reviewer #2. Thank you for your comments on our manuscript and for identifying the relevance of our study. For the different points of criticism you mentioned we presented a revised version with improvement of the information and formatting. Below you will find a point-by-point reply to your questions, indicating where the changes were made in the manuscript.

  • Line 21 to 24 - You mention some medical uses, such as assistance and rehabilitation. Here you could put some references. 

Reply: We added a reference to the review article dealing with these exoskeleton classes.

  • Line 27 to 28 - You mention some non-medical uses, like in soldiers, firefighters, etc. Here you could put some references.

Reply: We put a reference to the review article engaged in this exoskeleton class.

  • In the introduction you could mention the soft exoskeletons, and their advantages to be used in the support of the lower joints. You can be guided by the following article to further enrich your article:

 "Pérez Vidal, A. F., Rumbo Morales, J. Y., Ortiz Torres, G., Sorcia Vázquez, F. D. J., Cruz Rojas, A., Brizuela Mendoza, J. A., & Rodríguez Cerda, J. C. (2021, July). Soft exoskeletons: development, requirements, and challenges of the last decade. In Actuators (Vol. 10, No. 7, p. 166). MDPI".

Reply:  We added mention about soft exoskeletons in the introduction and referenced two reviews on this topic.

  • Table 1. Each paragraph must begin with a capital letter. You should also create a space between rows to differentiate them, because they get confused.

Reply:  We updated capitalization as suggested in the comment. We added top borders into table rows to differentiate them (as indicated in the Sensors template in Table 2).

  • Line 141 to 142 - You must start with a capital letter.

Reply: We updated bullet points so that each of them starts with a capital letter.

  • Line 153 and 187. It's confusing how you put this "Limitations" subtitle. Better put it as part of the paragraph. For example, "Limitations are..."

Reply:  We removed all occurrences of these subtitles.

Round 2

Reviewer 1 Report

Thanks to the author for responding to my comments. However, some minor problems still exist in this article. 

The discussion part still needs to be revised if possible. When I was reading, I thought some of the discussion section might expand in detail, but what you wrote jumped quickly to other parts and was not compact enough.

Author Response

Dear Reviewer. Thank you for your comment on our manuscript. We appreciate your comment and revised and re-structuralized the discussion section.